# Coupling of Thiazole-2-Amines with Isocyanide Ligands in *bis*-(Isocyanide) Platinum Complex: A New Type of Reactivity

**Yuliya A. Orekhova, Alexander S. Mikherdov, Vitalii V. Suslonov and Vadim P. Boyarskiy ***

Institute of Chemistry, Saint Petersburg State University, Saint Petersburg, 199034, Russia
* Correspondence: v.boiarskii@spbu.ru

**Abstract:** The treatment of *cis*-[PtCl$_2$(XylNC)$_2$] with thiazol-2-amines in a 2:1 ratio leads to a regioisomeric mixture of two binuclear complexes. These regioisomers are products of kinetic and thermodynamic control capable of regioisomerization. When the same reaction is carried out with a 5-fold excess of thiazol-2-amine, the nucleophile is able to react with the *in situ*-formed binuclear platinum(II) complexes, yielding a new type of *bis*-carbene platinum species. All new isolated compounds were characterized by $^1$H, $^{13}$C{$^1$H}, and $^{195}$Pt{$^1$H} NMR spectroscopy, high-resolution ESI-MS, and single-crystal X-ray diffraction.

**Keywords:** (Diaminocarbene)Pt$^{II}$ complexes; *bis*-(carbene)Pt$^{II}$ complex; regioisomerization; thiazole-2-amine; *bis*-(isocyanide) platinum complex

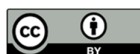

## 1. Introduction

Platinum metal complexes with diaminocarbene ligands (ADC—acyclic diaminocarbenes, NHC—N-heterocyclic carbenes) have become the subject of constant attention from modern researchers due to their successful application in catalysis [1–3], medicinal chemistry [4,5], and materials design [4]. Therefore, a large number of works have appeared that are devoted to methods for the synthesis of these species.

One such method is a platinum metal-mediated coupling of coordinated isocyanides with *N*-containing polynucleophiles. This reaction was first discovered by Chugaev more than 100 years ago [6] and studied by Belluco and coworkers in the 1970s [7–10]. These works have been systematized by Michelin, Pombeiro, and Guedes da Silva [11]. At the turn of the millennium, there was a surge in research in this area. It was associated with the development of catalysis of multicomponent reactions involving isocyanides (see, for example, reviews [12–15]) and with the discovery of good catalytic properties of acyclic diaminocarbene complexes (M-ADC) of platinum metals (see reviews and recent papers [2,3,16–29]) as well as their photophysical properties [30] and cytotoxicity [31]. It was found that the structure of the products for metal-mediated coupling strongly depends on the used nucleophile. M-ADCs are the main products when mono-*N*-nucleophiles are used as substrates. In the case of metal-mediated coupling of *bis*-isocyanide complexes with aromatic 2-amino-substituted azaheterocycles, binuclear complexes can form along with M-ADC [32–34].

Previously, we reported [35,36] that the treatment of *cis*-[PdCl$_2$(XylNC)$_2$] with thiazol-2-amine leads to a regioisomeric mixture of two binuclear complexes (Scheme 1), one of which proved to be kinetically controlled, while the generation of the other one is thermodynamically driven. In many instances, platinum(II) complexes exhibit similar reactivity as their palladium(II) congeners, although the former species are typically substantially more kinetically inert. This inertness is often used for the tapping and identification of those reaction intermediates that are elusive or quite unstable when

relevant palladium(II) species are employed. To get a deeper insight into the mechanism of the reported coupling, for this work we addressed the platinum(II) complex *cis*-[PtCl₂(XylNC)₂] and studied its reaction with thiazol-2-amines.

**Scheme 1.** The reactions of *cis*-[MCl₂(XylNC)₂] (M = Pt, Pd) with thiazol-2-amine.

## 2. Results and Discussion

### 2.1. The platinum(II)-Mediated Reaction

The reaction of *cis*-[PtCl₂(XylNC)₂] **1** with amino-substituted azaheterocycles **2** and **3** (2:1 molar ratio) in the presence of a base (*t*-BuONa) leads to regioisomeric mixtures of two platinum(II) binuclear diaminocarbene complexes **4a/b** and **5a/b** (Scheme 2, Route A); this coupling proceeds similarly to that of the corresponding palladium(II) complex *cis*-[PdCl₂(XylNC)₂]. The composition of the mixture depends on the reaction conditions. Thus, the reaction conducted at room temperature (RT) results almost exclusively in isomer **a** for complex **4** and in a 1:1 isomer mixture for complex **5**, whereas the quantity of isomer **b** increases upon reflux in 1,2-dichloroethane (Table 1). The product ratios in the reaction mixtures were monitored by ¹H NMR and determined by the peak integration.

**Scheme 2.** The studied reactions of **1**.

**Table 1.** Regioisomeric ratios obtained by the [1]H NMR integration of the reaction performed under different conditions.

| Mixture | Conditions | Ratio a:b for Pt | Ratio a:b for Pd [35,36] |
|---|---|---|---|
| **4a/b** | RT | 10:1 | 5:1 |
| | reflux | 1:3 | 1:4 |
| **5a/b** | RT | 1:1 | 4:1 |
| | reflux | 1:2 | 1:4 |

Refluxing the reaction mixture with an excess of **2** or **3** (3–10 equiv.) without an added base affords predominantly a new type of product: *bis*-carbene species **6** and **7** (Scheme 2, Route B). Such *bis*-carbene complexes of platinum(II) containing amino-azaheterocyclic fragments were not previously known. When the reaction is conducted at RT, it results in the generation of all three types of products, namely the two binuclear complexes (Scheme 2, top panel) and one mononuclear monoprotonated *bis*-carbene (bottom panel).

As in the case of binuclear palladium complexes [35], for platinum(II), the kinetically controlled regioisomer transforms over time into the thermodynamically controlled one. However, in contrast to the palladium congeners, the isomerization of **4a** under similar conditions (CDCl₃, 45 °C) took much longer: 110 days for the Pt(II) species vs. 14 days for the Pd(II) species (Figure 1).

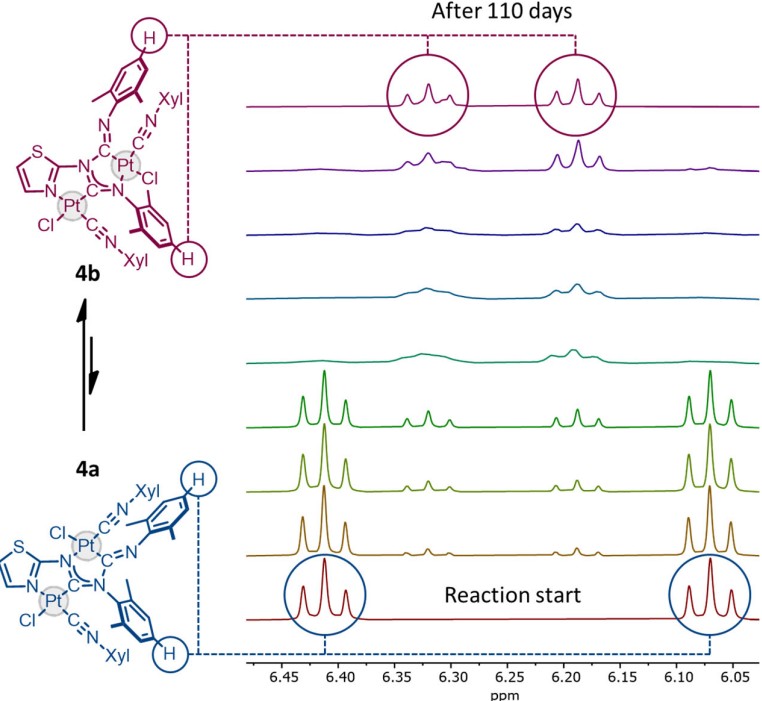

**Figure 1.** [1]H NMR monitoring of isomerization **4a** into **4b** in CDCl₃ at 45 °C.

For a deeper understanding of plausible reaction routes of the coupling of *bis*(isocyanide) Pt(II) complex with amino-substituted azaheterocycles, we studied the dependence of products ratio on reagents ratio in the model reaction of **1** and thiazole-2-amine **2** (Table 2). The experiments were conducted at RT for 12 h and the ratios were obtained by the [1]H NMR peak integration. In all reaction mixtures, irrespective of the concentration of **2**, we observed the formation of binuclear species **4a** and **4b**. Regioisomer **4b** is generated in trace amounts when the amount of **2** was less than one equiv.

Starting from a 1:1 ratio between **1** and **2**, we observed ¹H NMR resonances of *bis*-carbene **6**. Upon the increase of the starting concentration of **2**, the fraction of **6** also increases.

**Table 2.** Products ratio depending on the fraction of **2**.

| Ratio between Reactants 2:1 (Equiv.) | Conversion of 1 (%) | Product Ratios after 12 h (Equiv.) | | |
|---|---|---|---|---|
| | | **6** | **4a** | **4b** |
| 0.5 | 50 | 0 | 1.00 | trace |
| 1 | 70 | 0.09 | 0.91 | trace |
| 2 | 98 | 0.11 | 0.89 | 0 |
| 3 | 100 | 0.50 | 0.50 | 0 |
| 5 | 100 | 0.75 | 0.25 | 0 |
| 10 | 100 | 0.88 | 0.12 | 0 |

While studying the reaction mixture with the 5:1 ratio of **2** and **1**, we observed changes in the signals' intensities over time. Thus, the intensity of Me-groups' peaks from *bis*-carbene **6** increased, while those from binuclear species **4a** concurrently decreased. After two weeks at RT, we did not observe any signals from **4a**, while the reaction mixture remained homogeneous (Figure 2).

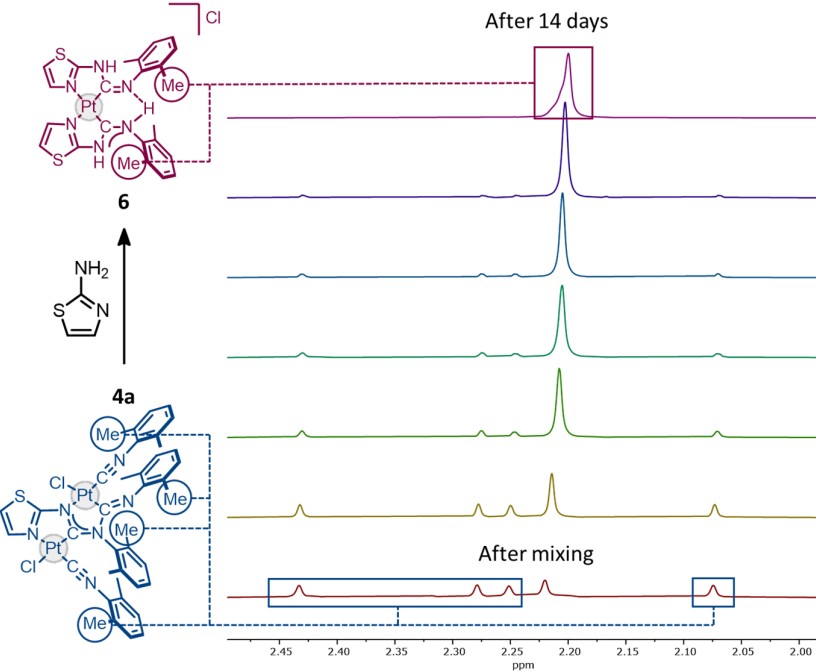

**Figure 2.** ¹H NMR monitoring of the reaction mixture with **1**/**2** ratio = 1:5 at RT.

We assumed that in an excess of **2**, binuclear species **4a** could transform into *bis*-carbene **6**. To prove this hypothesis, we performed the isolation of binuclear complex **4a** and then carried out its reaction with three equiv. of **2**. As expected, this reaction yielded *bis*-carbene **6**. Furthermore, we performed a cross-experiment comprising of the addition of three equiv. of another nucleophile, 5-methylthiazol-2-amine **3**, to complex **4a**. This experiment verified the generation of two *bis*-carbene complexes **7** and **8**, both identified in the reaction mixture by ¹H NMR and HRESI-MS (Scheme 3).

**Scheme 3.** The performed cross-experiment.

Considering the results of our experiments, we propose the following mechanism for the formation of **6** (Scheme 4). In the first stage, binuclear complex **4a** may proceed dissociation into cationic *bis*-isocyanide intermediate **i1** and anionic monocarbene complex **i2**, which can recombine into isomeric binuclear complex **4b**. However, in the presence of **2** acting as a nucleophile and a proton donor, each of these species can transform into monocarbene intermediate **i3**. The latter reacts with one more equiv. of thiazol-2-amine to eventually provide *bis*-carbene **6**.

**Scheme 4.** Postulated mechanism of the reaction.

### 2.2. Identification of **4a**, **4b**, **5a**, **6**, and **7**

Complexes **4a**, **4b**, **5a**, **6**, and **7** were isolated as pale-yellow crystals. After dissolution, all these species were characterized by HRESI-MS, 1D ($^1$H, $^{13}$C{$^1$H}, and $^{195}$Pt{$^1$H}), and 2D ($^1$H,$^1$H-COSY, $^1$H,$^1$HNOESY, $^1$H,$^{13}$C-HSQC, $^1$H,$^{13}$C-HMBC) NMR spectroscopy. Complex **5b** was not isolated in pure form and was characterized in the mixture with the other regioisomer by HRESI-MS and $^1$H NMR.

The $^1$H NMR spectra of **4**–**5a/b** in each case display signals for each of the four Xyl fragments as well as proton signals of the thiazole core. In the $^{13}$C{$^1$H} NMR spectra of **4**–**5a/b**, two distinct resonances of two Pt-bound NCN fragments were observed at

154.07–154.86 ppm and 182.20−184.50 ppm for kinetically controlled isomers **4**–**5a** and at 149.98 and 170.83 ppm for thermodynamically controlled isomer **4b**. The attribution of these signals was performed by 2D (HMBC, HSQC) NMR, and the position of the signals is similar to the analogous binuclear Pd(II) species [35,36]. Finally, in the $^{195}$Pt{$^1$H} spectra of binuclear complexes **4**−**5a/b**, two signals from two Pt(II) centers were observed in the range from −3750 to −3820 ppm.

At the same time, the $^1$H NMR spectra of complexes **6** and **7** exhibit only one set of signals for both two xylyl and two thiazole fragments, indicating the symmetric structure of complexes. Similarly, the $^{13}$C{$^1$H} NMR spectra of the complexes display only one signal corresponding to the carbene carbon atom at ca. 160 ppm, and the $^{195}$Pt{$^1$H} NMR spectra display only one signal around −3850 ppm. One should also mention the presence in the $^1$H NMR spectra of a high-field signal (12.5–14.0 ppm), corresponding to the N···H···N hydrogen atom involved in a resonance-assisted hydrogen bond [37] between two carbene ligands.

The solid-state structures of **4**–**5a**,**b**, **6**, and **7** were established by single-crystal X-ray diffraction (XRD). The plot of the XRD structures is shown in Figure 3, while the crystal data, data collection parameters, and structure refinement data are given in Tables S1 and S2.

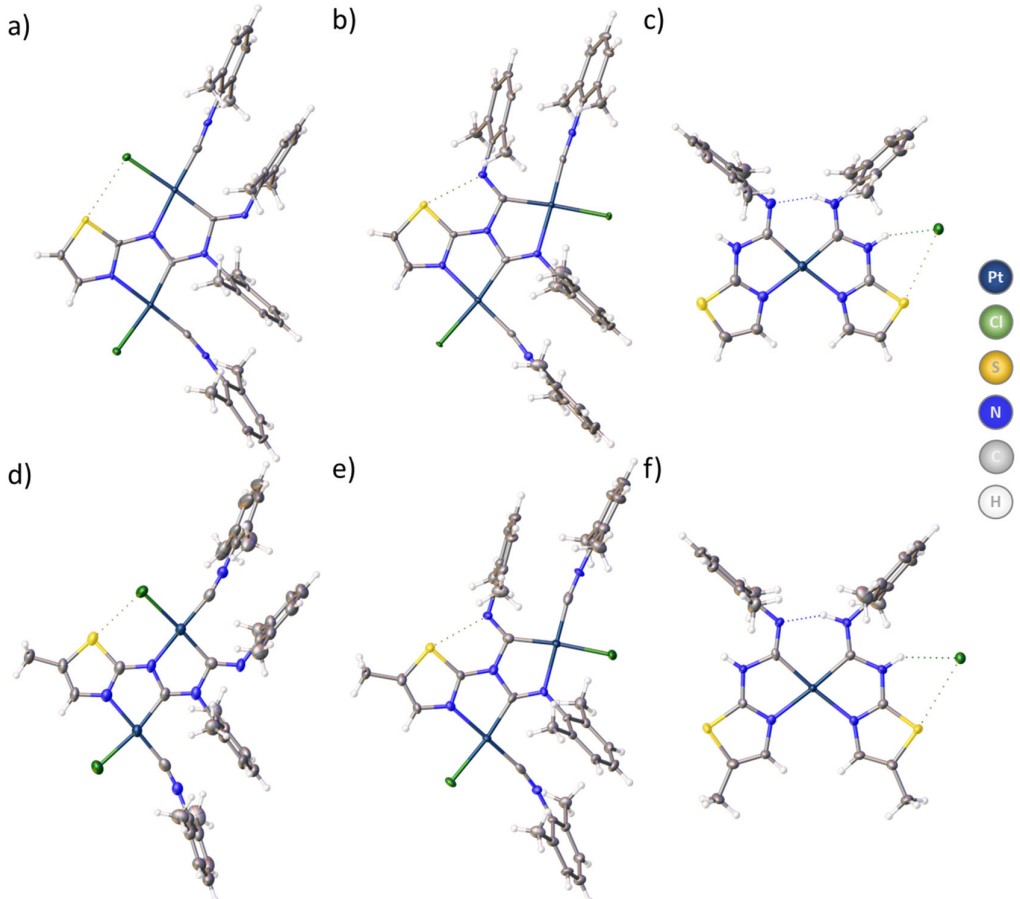

**Figure 3.** Views of crystal structures of complexes: **4a** (**a**), **4b** (**b**), **6** (**c**), **5a** (**d**), **5b** (**e**), **7** (**f**). Dotted lines indicate S···Cl, S···N, H···Cl and H···N noncovalent interactions.

The structures of complexes **4**–**5a/b** are similar to those for the relevant Pd(II) binuclear complexes with thiazole fragments [35,36]. In structures of **4**–**5a/b**, both metal centers adopt a slightly distorted square planar geometry, and the isocyanide ligands are in

the cis-position to the NCN fragments. The bond lengths of the two coordinated CN groups fall in the interval of 1.139–1.165 Å which is typical for the common range of the CN triple bonds in the related isocyanide Pd(II) and Pt(II) complexes [35,36]. In **4–5a**, the lengths of the C−N bonds in the one NCN fragment connected to Pt are close (1.336(8)–1.373(8) Å), and they are intermediate between the typical double (1.260(9) Å) and single bonds (1.452(8) Å) [38]. In the other NCN fragment, one C–N bond is closer to single (1.454(7)–1.468(7) Å), while the other bond is a typical double bond (1.274–1.276 Å). For **4–5b**, one C−N bond in each NCN fragment is double (1.267–1.308 Å), and the other one is closer to a single bond (1.400–1.434 Å).

In the case of complexes **6** and **7**, both carbene fragments are located in *cis*-position. Despite the single set of signals in the solution NMR spectra for two chelate carbene ligands, in a solid state, two carbene N-C-N fragments appear to be different for both complexes. The first N-C-N fragment possesses both N atoms protonated, and the lengths of both C−N bonds are intermediate between the typical double and single bonds (1.304–1.379 Å), reflecting the diaminocarbene nature. In the second N-C-N fragment, only one N atom is protonated, whereas the second N atom is involved in the intramolecular hydrogen bonding. The latter results in different C−N bond lengths in the fragment: the bond with the protonated N atom is single (1.271–1.283 Å) while the other one is closer to a double bond (1.409–1.412 Å). The disagreement between XRD and NMR data can be explained by the fast exchange of the N-H proton between two N-C-N fragments in a solution, whereas in a solid state, this proton is attributed only to one nitrogen atom.

All other bond lengths in the complexes are typical, and their values agree with those reported for related Pt(II) and Pd(II) carbene and isocyanide complexes [26,32,34–36].

### 2.3. Noncovalent Interactions in the Structures of *4a, 4b, 5a, 6,* and *7*

Inspection of the crystallographic data for the obtained complexes suggests the presence of intermolecular and intramolecular noncovalent interactions in their structures (dotted lines in Figure 3). Like previously reported Pd$^{II}$ species [35], the binuclear Pt$^{II}$ complexes display the presence of intermolecular chalcogen bonds [39]: S···Cl interactions in the structures of **4a–b** and S···N interactions in the structures of **5a–b**. At the same time, both bis-carbene complexes **7** and **8** feature intermolecular bifurcated chalcogen-hydrogen bonds $\mu_{(S,N-H)}$Cl [34] with the chloride anion in outer coordination sphere, as well as intramolecular N–H···N hydrogen bonds between two carbene fragments. To confirm the presence of these noncovalent interactions in the crystal structures and quantify their energies from a theoretical point of view, we carried out DFT calculations along with topological analysis of the electron density distribution (quantum theory of atoms in molecules, QTAIM) [40] for the XRD geometries of complexes. The results of the QTAIM analysis are visualized in Figure 4 and summarized in Table 3.

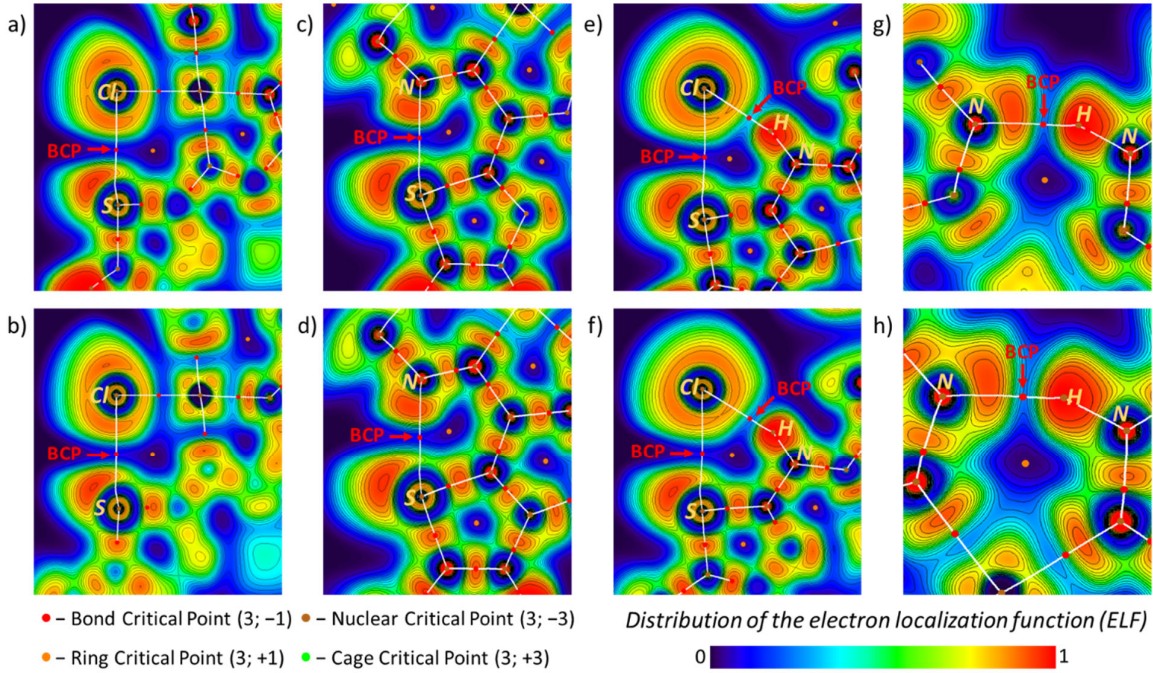

- • – Bond Critical Point (3; –1)   • – Nuclear Critical Point (3; –3)
- • – Ring Critical Point (3; +1)   • – Cage Critical Point (3; +3)

*Distribution of the electron localization function (ELF)*

0      1

**Figure 4.** Visualization of the electron localization function (ELF), QTAIM bond critical point (BCP), and bond paths (white lines) corresponding S···Cl chalcogen bonds in structures of **4a** (**a**) and **4b** (**b**), S···N chalcogen bonds in structures of **5a** (**c**) and **5b** (**d**), bifurcated chalcogen-hydrogen bonds $\mu_{(S,N-H)}$Cl in structures of **6** (**e**) and **7** (**f**), and N–H···N hydrogen bonds in structures of **6** (**g**) and **7** (**h**).

**Table 3.** Noncovalent interactions in the structures of complexes: geometrical parameters of noncovalent interactions (Å and deg), values of the density of all electrons multiplied on $\text{sign}(\lambda_2)$—$\text{sign}(\lambda_2)\varrho(r)$, Laplacian of electron density—$\nabla^2\varrho(r)$, energy density—$H(r)$, potential energy density—$V(r)$, and Lagrangian kinetic energy—$G(r)$ (a.u.) at the BCPs (3, –1), energies of noncovalent interactions defined by two approaches.

| Complex | Contact | $d$(D···A) | $\angle$(R-D···A) | $\text{sign}(\lambda_2)\varrho$ | $\nabla^2\varrho(r)$ | $H(r)$ | $V(r)$ | $G(r)$ | $E_{int}$ [a] | $E_{int}$ [b] |
|---|---|---|---|---|---|---|---|---|---|---|
| **4a** | S···Cl | 3.075 | 172.01 | -0.018 | 0.046 | 0.0004 | -0.011 | 0.011 | 3.4 | 3.0 |
| **5a** | S···Cl | 3.072 | 164.88 | -0.018 | 0.047 | 0.0003 | -0.011 | 0.011 | 3.5 | 3.1 |
| **4b** | S···N | 2.696 | 160.17 | -0.025 | 0.078 | 0.0019 | -0.016 | 0.018 | 4.9 | 4.7 |
| **5b** | S···N | 2.716 | 160.42 | -0.024 | 0.075 | 0.0019 | -0.015 | 0.017 | 4.7 | 4.5 |
| **6** | S···Cl | 3.417 | 170.15 | -0.011 | 0.023 | 0.0003 | -0.005 | 0.006 | 1.6 | 1.5 |
| | H···Cl | 1.976 | 160.36 | -0.045 | 0.068 | -0.0052 | -0.027 | 0.022 | 8.6 | 6.0 |
| | H···N | 1.707 | 151.39 | -0.052 | 0.144 | -0.0035 | -0.043 | 0.039 | 13.5 | 10.6 |
| **7** | S···Cl | 3.331 | 172.26 | -0.013 | 0.027 | 0.0002 | -0.006 | 0.006 | 1.9 | 1.7 |
| | H···Cl | 2.163 | 154.27 | -0.032 | 0.051 | -0.0024 | -0.017 | 0.015 | 5.5 | 4.0 |
| | H···N | 1.680 | 152.56 | -0.056 | 0.144 | -0.0055 | -0.047 | 0.042 | 14.8 | 11.2 |

[a] $E_{int} = -V(r)/2$ [41]; [b] $E_{int} = 0.429G(r)$ [42].

The QTAIM analysis demonstrates the presence of appropriate bond critical points (3, –1) (BCPs) for all discussed interactions. The observed low magnitude of the electron density, positive values of the Laplacian, and close to zero energy density in the BCPs are typical for noncovalent interactions. We have defined energies for the studied contacts in accordance with the conventional approach by Espinosa et al. [41] and Vener et al. [42]. For the binuclear Pt[II] complexes **4–5a,b**, the energies of the intramolecular S···Cl and S···N chalcogen bonds are in the same range as for Pd[II] analogs (3–5 kcal/mol). Note also the intramolecular N–H···N hydrogen bonds of medium strength (10–15 kcal/mol) in the

structures of bis-carbene complexes **6** and **7**. The latter interaction should also contribute to stabilization of these new types of complexes.

## 3. Materials and Methods

### 3.1. General

The complex *cis*-[PtCl$_2$(CNXyl)$_2$] (complex **1**) was synthesized using the reported procedure [43]. All reagents and solvents were obtained from commercial sources and used as received, apart from chloroform, which was dried by conventional distillation over calcium chloride. Mass spectra were obtained on a Bruker micrOTOF spectrometer equipped with an electrospray ionization (ESI) source, and a mixture of MeOH and CH$_2$Cl$_2$ was used for sample dissolution. The instrument was operated in a positive ion mode using a *m/z* range of 50–3000 with the capillary voltage of the ion source set at −4500 V and the capillary exit at 50–150 V. The most intensive peak in the isotopic pattern is reported. The 1D ($^1$H, $^{13}$C, $^{195}$Pt) NMR spectra were acquired on a Bruker Avance 400 spectrometer, whereas the 2D ($^1$H,$^1$H-COSY, $^1$H,$^1$H-NOESY, $^1$H,$^{13}$C-HSQC, and $^1$H,$^{13}$C-HMBC) NMR correlation experiments were recorded on a Bruker Avance III 500 MHz spectrometer. All NMR spectra were measured in CDCl$_3$ at ambient temperature.

### 3.2. Synthesis of the Complexes

**Complexes 4a and 5a**. A solution of complex **1** (0.04 mmol, 20 mg), NaOBu$^t$ (0.04 mmol, 5.5 mg), and nucleophile **2** (**3**) (0.02 mmol) in 5 mL of dichloromethane placed in a 10 mL vial were stirred under ultrasonication for 40 min at room temperature. The resulting mixture was filtered off from some insoluble material and then dried under low pressure.

Yield 70%. HR ESI$^+$-MS: found *m/z* 1046.1827 [M–Cl]$^+$, calculated C$_{39}$H$_{38}$N$_6$ClSPt$_2$$^+$: 1046.1842. NMR $^1$H: 7.97 (d, 1H, H$^3$, thiazole, J = 4.07), 7.18 (t, 1H, H$^4$, Xyl-A, J = 7.48), 7.09 (t, 1H, H$^4$, Xyl-D, J = 7.48), 7.04 (d, 2H, H$^{3,5}$, Xyl-A, J = 7.64), 6.95 (d, 2H, H$^{3,5}$, Xyl-D, J = 7.63), 6.93 (d, 1H, H$^2$, thiazole, J = 3.89), 6.87 (d, 2H, H$^{3,5}$, Xyl-B, J = 7.59), 6.59 (d, 2H, H$^{3,5}$, Xyl-C, J = 7.52), 6.39 (t, 1H, H$^4$, Xyl-B, J = 7.56), 6.05 (t, 1H, H$^4$, Xyl-C, J = 7.50), 2.42 (s, 6H, Me, Xyl-B), 2.27 (s, 6H, Me, Xyl-A), 2.24 (s, 6H, Me, Xyl-D), 2.06 (s, 6H, Me, Xyl-C). NMR $^{13}$C: 183.93 (C$^1$, thiazole), 183.19 (C$^4$), 154.73 (C$^5$), 150.55 (C$^1$, Xyl-C), 142.16 (C$^1$, Xyl-B), 136.51 (2C, C$^{2,6}$, Xyl-B), 134.50 (2C, C$^{2,6}$, Xyl-D), 134.38 (2C, C$^{2,6}$, Xyl-A), 132.45 (C$^2$, thiazole), 129.69 (C$^4$, Xyl-B), 129.52 (C$^4$, Xyl-A), 128.87 (C$^4$, Xyl-D), 127.87 (4C, C$^{3,5}$, Xyl-A, Xyl-B), 127.70 (2C, C$^{3,5}$, Xyl-C), 127.45 (2C, C$^{3,5}$, Xyl-D), 126.54 (2C, C$^{2,6}$, Xyl-C), 123.30 (C$^4$, Xyl-C), 112.54 (C$^3$, thiazole), 19.64 (2C, Me, Xyl-C), 19.12 (2C, Me, Xyl-B), 18.64 (2C, Me, Xyl-A), 18.56 (2C, Me, Xyl-D). NMR $^{195}$Pt: −3809.44, −3817.46.

**5a**. Yield 96%. HR ESI⁺-MS: found *m/z* 1060.1982 [M–Cl]⁺, calculated $C_{40}H_{40}N_6ClSPt_2^+$: 1060.1993 NMR ¹H (400 MHz, CDCl₃): 7.64 (s, 1H, H², thiazole), 7.18 (t, 1H, H⁴, Xyl-A, *J* = 7.7), 7.09 (t, 1H, H⁴, Xyl-D, *J* = 7.60), 7.04 (d, 2H, H³,⁵, Xyl-A, *J*= 7.60), 6,94 (d, 2H, H³,⁵, Xyl-D, *J* = 7.70), 6.86 (d, 2H, H³,⁵, Xyl-B, *J* = 7.59), 6.58 (d, 2H, H³,⁵, Xyl-C, *J* = 7.50), 6.38 (t, 1H, H⁴, Xyl-B, *J* = 7.50), 6.04 (t, 1H, H⁴, Xyl-C, *J* = 7.50), 2.41 (s, 6H, Me, Xyl-B), 2.38 (s, 3H, Me, thiazole) 2.26 (s, 6H, Me, Xyl-A), 2.24 (s, 6H, Me, Xyl-D), 2.06 (s, 6H, Me, Xyl-C). NMR ¹³C (101 MHz, CDCl₃): 183.04 (C¹, thiazole). 182.39 (C⁴), 150.59 (C¹, Xyl-C), 154.07 (C⁵), 142.21 (C¹, Xyl-B), 136.54 (2C, C²,⁶, Xyl-B), 134.51 (2C, C²,⁶, Xyl-D), 134.38 (2C, C²,⁶, Xyl-A), 129.61 (C⁴, Xyl-B), 129.47 (C⁴, Xyl-A), 128.83 (C⁴, Xyl-D), 127.78 (4C, C³,⁵, Xyl-A, Xyl-B), 127.68 (2C, C³,⁵, Xyl-C), 127.43 (2C, C³,⁵, Xyl-D), 126.57 (C², thiazole), 126.48 (2C, C²,⁶, Xyl-C), 123.24 (C⁴, Xyl-C), 19.63 (2C, Me, Xyl-C), 19.12 (2C, Me, Xyl-B), 18.64 (2C, Me, Xyl-A), 18.56 (2C, Me, Xyl-D), 17.74 (Me, thiazole). NMR ¹⁹⁵Pt (86 MHz, CDCl₃): −3807.07, −3812.47.

**Complex 4b**. A solution of complex **1** (0.04 mmol, 20 mg), NaOBuᵗ (0.04 mmol, 5.5 mg) and nucleophile **2** (0.02 mmol, 1.9 mg) in 5 mL of dichloroethane placed in a 10 mL vial was stirred at 70 °C for 2 days. The resulting mixture was filtered off from some insoluble material and then dried under low pressure. The title product was isolated by slow crystallization out of the mixture of dichloromethane and acetone (1:2) under normal pressure and room temperature. The crystals were separated from the solution (0.5–1 mL) and dried under low pressure at room temperature.

**4b**. Yield 38%. HR ESI⁺-MS: found *m/z* 1046.1827 [M–Cl]⁺, calculated for $C_{39}H_{38}N_6ClSPt_2^+$: 1046.1842. NMR ¹H: 8.18 (d, 1H, H³, thiazol, *J* = 4.0), 7.20 (t, 1H, H⁴, Xyl-A, *J* = 7.5), 7.11 (t, 1H, H⁴, Xyl-D, *J* = 7.5), 7.07 (d, 2H, H³,⁵, Xyl-A, *J* = 7.6), 7.00 (d, 1H, H², thiazol, *J*= 4.0), 6.96 (d, 2H, H³,⁵, Xyl-D, *J* = 7.6), 6.87 (d, 2H, H³,⁵, Xyl-B, *J* = 7.5), 6.76 (d, 2H, H³,⁵, Xyl-C, *J* = 7.5), 6.32 (t, 1H, H⁴, Xyl-B, *J* = 7.5), 6.19 (t, 1H, H⁴, Xyl-C, *J* = 7.5), 2.44 (s, 6H, Me, Xyl-B), 2.31 (s, 6H, Me, Xyl-A), 2.29 (s, 6H, Me, Xyl-C), 2.24 (s, 6H, Me, Xyl-D). NMR ¹³C: 170.83 (C⁴), 164.37 (C¹, thiazole), 149.98 (C⁵), 148.81 (C¹, Xyl-C), 146.94 (C¹, Xyl-B), 134.58 (2C, C²,⁶, Xyl-D), 134.37 (2C, C²,⁶, Xyl-A), 133.41 (2C, C²,⁶, Xyl-B), 132.53 (C², thiazole), 129.34 (C⁴, Xyl-A), 128.75 (C⁴, Xyl-D), 128.38 (2C, C²,⁶, Xyl-C), 127.78 (C⁴, Xyl-B), 127.69 (2C, C³,⁵, Xyl-C), 127.63 (2C, C³,⁵, Xyl-A), 127.33 (2C, C³,⁵, Xyl-B), 127.26 (2C, C³,⁵, Xyl-D), 123.51 (C⁴, Xyl-C), 113.16 (C³, thiazole), 19.39 (2C, Me, Xyl-C), 19.36 (2C, Me, Xyl-B), 18.52 (2C, Me, Xyl-D), 18.46 (2C, Me, Xyl-A). NMR ¹⁹⁵Pt: −3752.35, −3779.41.

**Complexes 6 and 7**. A solution of complex **1** (0.04 mmol, 20 mg), NaOBuᵗ (0.04 mmol, 5.5 mg), and nucleophile **2** (**3**) (0.11 mmol) in 5 mL of dichloroethane placed in a 10 mL vial was stirred at 70 °C for 2 days. The resulting mixture was filtered off from some

insoluble material and concentrated to 1 mL under low pressure and purified from residual binuclear complexes **4** (**5**) via column chromatography with dichloromethane (19 mL) as eluent on silica gel as a sorbent. The title product was washed from the silica gel with methanol (2 × 2 mL) and dried under low pressure.

**6.** Yield 47%. HR ESI$^+$-MS: found *m/z*: 656.1230 [M–Cl]$^+$, calculated for $C_{24}H_{25}N_6PtS_2^+$: 656.1238. NMR $^1$H: 13.92 (s, 1H, H$^1$), 7.51 (d, *J* = 3.9 Hz, 2H, H$^2$ thiazol), 7.16–7.09 (m, 6H, Xyl), 7.03 (d, *J* = 3.9 Hz, 2H, H$^3$, thiazol), 2.22 (s, 12H, Me). The signals of the remaining NH protons were not detected, presumably due to exchange. NMR $^{13}$C: 173.17 (2C, C$^1$, thiazol), 160.63 (2C, C$^4$), 139.14 (2C, C$^1$, Xyl), 135.19 (2C, C$^2$, thiazol), 131.67 (4C, C$^{2,6}$), 128.64 (4C, C$^{3,5}$, Xyl), 126.65 (2C, C$^4$, Xyl), 113.01 (2C, C$^3$, thiazol), 18.30 (4C, Me, Xyl). NMR $^{195}$Pt: −3853.80.

**7**. Yield 43%. HR ESI$^+$-MS: found *m/z* 684.1525 [M–Cl]$^+$, calculated $C_{26}H_{29}N_6S_2Pt^+$: 684.1538. NMR $^1$H (400 MHz, CDCl$_3$): 12.60 (s, 1H, H$^1$), 7.09–6.96 (m, 8H, Xyl, C$^3$, thiazole), 2.39 (s, 6H, Me, thiazole), 2.21 (s, 12H, Me, Xyl). The signals of the remaining NH protons were not detected, presumably due to exchange. NMR $^{13}$C (101 MHz, CDCl$_3$): 174.57 (2C, C$^1$, thiazole), 160.18 (2C, C$^4$), 141.63 (2C, C$^1$, Xyl), 131.76 (4C, C$^{2,6}$, Xyl), 128.22 (4C, C$^{3,5}$, Xyl), 125.39 (2C, C$^4$, Xyl), 124.31 (1C, C$^3$, thiazole), 18.72 (4C, Me, Xyl), 12.80 (1C, Me, thiazole). NMR $^{195}$Pt (86 MHz, CDCl$_3$): −3850.38.

### 3.3. X-Ray Diffraction

The SC-XRD experiments were carried out using Oxford Diffraction "Xcalibur" (**4a,b**, **6**), Rigaku "SuperNova XtaLAB" (**5a**, **5b**), and Rigaku "Synergy XtaLAB" (**7**) diffractometers with monochromated MoK$\alpha$ and CuK$\alpha$ radiation. All crystals were kept at 100(2) K during data collection. The crystal structures were solved using ShelXT [44] and Superflip [45–47] structure solution programs and refined by means of ShelXL [48] incorporated in the Olex2 [49] program complex. All crystallographic data are available free of charge via the Cambridge Crystallographic Data Centre (CCDC 2212275–2212279, accessed on 11 October 2022; 2214271, accessed on 20 October 2022; https://www.ccdc.cam.ac.uk/structures/).

### 3.4. Computational Details

For studies of the nature of noncovalent interactions, the single-point calculations based on the experimental X-ray geometries for wavefunction generation were carried

out for **4–5a**,**b**, while for the structures of **6** and **7**, the positions of the heavy atoms were fixed and only the positions of H atoms were optimized. All calculations were carried out at the DFT level of theory using the M06 [50] functional with the empirical corrections for dispersion in accordance with Grimme's D3 model [51] and def2-SVP basis [52,53] set for all atoms by Gaussian-16 program package [54]. The Hessian matrices were calculated analytically for all optimized model structures to prove the location of correct minima on the potential energy surface (no imaginary frequencies were found). The QTIAM analysis was carried out using Multiwfn 3.8 software [55]. The Cartesian atomic coordinates for model structures are given as XYZ files in Supplementary Files.

## 4. Conclusions

In this work, we have shown that the reaction of bis(isocyanide) complexes of platinum(II) with aminoazaheterocyles can lead not only to the mixture of binuclear regioisomeric complexes but also to a new type of *bis*-carbene species not observed previously for the reaction on Pd$^{II}$ metal center. The formation of the *bis*-carbene complexes occurs via the reaction of binuclear diaminocarbene Pt$^{II}$ species with aminothiazoles, and this process was studied experimentally by $^1$H NMR and HR-ESIMS methods. The isolated compounds were characterized by $^1$H, $^{13}$C{$^1$H}, and $^{195}$Pt{$^1$H} NMR spectroscopy and HRESI-MS in a solution, whereas the solid-state structures of all complexes were elucidated by single-crystal XRD. In addition, we observed several types of noncovalent interactions in the crystal structures of obtained complexes (S···Cl/N chalcogen bonds, bifurcated chalcogen-hydrogen bonds $\mu_{(S,N-H)}$Cl, N–H···N hydrogen bonds), and their presence was confirmed by the theoretical studies.

**Supplementary Materials:** The following supporting information can be downloaded at: https://www.mdpi.com/article/10.3390/inorganics10120221/s1, Figures S1–S15: NMR $^1$H, $^{13}$C, and $^{195}$Pt spectra of complexes **4–7**; Tables S1 and S2: crystal data and structure refinement for complexes **4–7**; crystallographic information files (CIF) and checkCIF report files for complexes **4–7**.

**Author Contributions:** Conceptualization, V.P.B.; methodology, A.S.M.; resources, V.V.S.; investigation, Y.A.O. and A.S.M.; writing—original draft preparation, A.S.M.; writing—review and editing, V.P.B.; visualization, V.V.S.; supervision, V.P.B. All authors have read and agreed to the published version of the manuscript.

**Funding:** This research was funded by the Russian Science Foundation, grant number 19-13-00008.

**Institutional Review Board Statement:** Not applicable.

**Informed Consent Statement:** Not applicable.

**Data Availability Statement:** The data presented in this study are available on request from the corresponding author.

**Acknowledgments:** Physicochemical studies were performed at the Center for Magnetic Resonance, the Center for X-ray Diffraction Studies, and the Center for Chemical Analysis and Materials Research of Saint Petersburg State University.

**Conflicts of Interest:** The authors declare no conflict of interest. The funders had no role in the design of the study; in the collection, analyses, or interpretation of data; in the writing of the manuscript; or in the decision to publish the results.

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
