# Peer review of "Coupling of Thiazole-2-Amines with Isocyanide Ligands in bis-(Isocyanide) Platinum Complex: A New Type of Reactivity"

_inorganics, doi:10.3390/inorganics10120221_

Round 1

Reviewer 1 Report

This manuscript describes some interesting results concerning the reaction of a platinum(II) bis-isocyanide complex with 2-amino-substituted thiazoles, leading to the formation of diaminocarbene derivatives or related species. Similar palladium(II) have already been described by the authors, except from a new type of bis-carbene species.

I recommend publication of this manuscript in Inorganics, once the following minor points are taken into consideration by the authors:

Results and Discussion:

Line 53: aminoazaheterocycle 2 and 3, more appropriate seems: amino-substituted azaheterocycles 2 and 3 (and throughout the manuscript).

Lines 58 and 59: at RT exclusively gave isomer a..this seems not to be in accordance with Table 1 (for 5a/b a 1:1 mixture is noted). 

Table 1. Put 4a/b instead of a/b (4 is missing).

Figure 1. The two xylyl groups originating both triplet signals (para hydrogens) should be identified in the figure or in the caption (or in both).

Table 2. The reaction time should be indicated.

Lines 161,162: For the structure of 4,5b there is a mistake in the bond length comment for the NCN fragments. Put double instead of single and single instead of double.

Figure 3: b) and c) notations are missing.

Author Response

Comments and Suggestions for Authors

This manuscript describes some interesting results concerning the reaction of a platinum(II) bis-isocyanide complex with 2-amino-substituted thiazoles, leading to the formation of diaminocarbene derivatives or related species. Similar palladium(II) have already been described by the authors, except from a new type of bis-carbene species.

I recommend publication of this manuscript in Inorganics.

 Thank you for your appreciation of our work and for your careful reading, which helped improve the manuscript.

 Once the following minor points are taken into consideration by the authors:

Results and Discussion:

  1. Line 53: aminoazaheterocycle 2 and 3, more appropriate seems: amino-substituted azaheterocycles 2 and 3 (and throughout the manuscript).

Done

  1. Lines 58 and 59: at RT exclusively gave isomer a..this seems not to be in accordance with Table 1 (for 5a/b a 1:1 mixture is noted).

We have corrected the text. Now it looks like this:

“at room temperature (RT) results almost exclusively in isomer a for complex 4 and in a 1:1 isomer mixture for complex 5”.

  1. Table 1. Put 4a/b instead of a/b (4 is missing).

Done

  1. Figure 1. The two xylyl groups originating both triplet signals (para hydrogens) should be identified in the figure or in the caption (or in both).

We marked on the structures to the left of the spectra the protons whose signals were used for monitoring. Similarly, we marked the monitored methyl groups in Figure 2.

  1. Table 2. The reaction time should be indicated.

These experiments were conducted for 12 h. The information was added to the table and to the text. Now it looks like this:

“The experiments were conducted at RT for 12 h”.

  1. Lines 161,162: For the structure of 4,5b there is a mistake in the bond length comment for the NCN fragments. Put double instead of single and single instead of double.

Done

  1. Figure 3: b) and c) notations are missing.

Done

Reviewer 2 Report

The authors of the manuscript "Coupling of Thiazole-2-Amines with Isocyanide Ligands in bis-(Isocyanide) Platinum Complex: A New Type of Reactivity" studied reactions of bis(isocyanide) complexes of platinum(II) with 2-aminothiazoles. It was established that, depending on the conditions, not only the expected binuclear complexes, but also new type bis-carbene platinum species are formed. The synthesized compounds are well characterized by spectral methods and single-crystal X-ray diffraction data. The manuscript contains sufficient scientific novelty and may be published in Inorganics.

Author Response

Comments and Suggestions for Authors

The authors of the manuscript "Coupling of Thiazole-2-Amines with Isocyanide Ligands in bis-(Isocyanide) Platinum Complex: A New Type of Reactivity" studied reactions of bis(isocyanide) complexes of platinum(II) with 2-aminothiazoles. It was established that, depending on the conditions, not only the expected binuclear complexes, but also new type bis-carbene platinum species are formed. The synthesized compounds are well characterized by spectral methods and single-crystal X-ray diffraction data. The manuscript contains sufficient scientific novelty and may be published in Inorganics.

Thank you for your high appreciation of our work.